# Entropic Divergence and Entropy Related to Nonlinear Master Equations

**DOI:** 10.3390/e21100993

**Published:** 2019-10-11

**Authors:** Tamás Sándor Biró, Zoltán Néda, András Telcs

**Affiliations:** 1Wigner Research Centre for Physics, 1121 Budapest, Hungary; Telcs.Andras@wigner.mta.hu; 2Department of Physics, Babeş-Bolyai University, 400084 Cluj-Napoca, Romania; zneda@phys.ubbcluj.ro; 3Department of Computer Science and Information Theory, Budapest University of Technology and Economics, 1111 Budapest, Hungary; 4Department of Quantitative Methods, University of Pannonia, 8200 Veszprém, Hungary; 5VIAS Virtual Institute for Advanced Studies, 1039 Budapest, Hungary

**Keywords:** entropy, entropic divergence, master equation, reset, preferential growth

## Abstract

We reverse engineer entropy formulas from entropic divergence, optimized to given classes of probability distribution function (PDF) evolution dynamical equation. For linear dynamics of the distribution function, the traditional Kullback–Leibler formula follows from using the logarithm function in the Csiszár’s f-divergence construction, while for nonlinear master equations more general formulas emerge. As applications, we review a local growth and global reset (LGGR) model for citation distributions, income distribution models and hadron number fluctuations in high energy collisions.

## 1. Motivation

The entropic convergence is being intensively studied alongside functional inequalities.

The log–Sobolev inequality and a family of other related functional inequalities have been developed to control the speed of convergence of stochastic processes on the Euclidian space and on manifolds. The control is given in a form of an upper bound on the speed of convergence in one or other metrics, e.g., Wasserstein distance, Lp-norm or other. The most important development by this approach is the incorporation of the curvature condition, which replaces the condition on the lower bound of the spectral gap. The latter one can be typically used on compact sets while the former allows us to control the convergence on non-compact sets, on spaces of infinite diameter or measure.

The theory of measure concentration (“entropic distance shrinking”) received a new impact from the work of Marton [1] and Talagrand [2]. This initiative later grow into a unified treatment by Otto and Villanyi [3] incorporating mass transport theory and gradient flow on manifolds of measures. All those efforts are focused on finding conditions for the convergence of the process under scrutiny. In these works several extensions of the log-Sobolev inequality, Wasserstein distance and Fisher–Donsker–Varadhan information is established [4].

The present work is dealing with processes for which such methods does not work to our best knowledge. Convergence is ensured in a different way. A metric, in particular a Kullback-type divergence [5] is created in which—thanks to the intimate relation to the structure of the master equation—it can be shown that its time derivative is negative.

For recent developments dealing with discrete Markov chains see also References [6,7] and references therein.

The classical entropy—probability relation, pioneered by Boltzmann, used and discussed by Gibbs, Planck and Shannon [8,9,10,11,12] (just to mention a few), is under steady generalization attempts, e.g., [13,14,15,16,17,18]. Most of the suggestions are mathematically motivated, formulated in terms of axioms (Khintchine or other) [19,20], sometimes abandoning the additivity property for factorizing probabilities, therefore treating statistical independence in an unorthodox way [21]. The motivation behind doing so lies in the study of complex systems, where complexity reveals itself in long-range correlations causing unusual behavior in the thermodynamical limit. It is common in such phenomena that interface effects, multiplied with a characteristic correlation length, are not smaller than bulk (volume) effects. This ruins the familiar hierarchy valid for big systems in the classical thermodynamics and is mathematically reflected in the non-extensivity property. Hence formulas generalizing the logarithmic function at the core of the entropy formula are necessarily non-extensive. Besides this, some approaches keep the logarithm but use a function of the probability in the formula. For example, in the field of sensor data analysis, the Deng entropy is in use [22,23,24].

By the generalized entropy formula, important properties, like non-negativity, expansibility, convexity are still to be satisfied [25]. It is, however, hard to convince about one or the other “next to simplest” generalization without investigating physical phenomena which realize or—if we are lucky—even suggest preferring a given formula against others [26,27]. First, in a deductive way someone suggests a formula and then checks its fundamental properties before applying to a number of physical observations. Any suggested new entropy formula must include the conventional one as a limiting case, since this has already the largest observational support. A second way to select a favorite formula may rely on starting with the study of a given class of complex system behavior and adjust the classical definition accordingly. Most importantly, one expects that the formula commands the entropy to increase in a closed system, under the given microscopic dynamics already known or assumed.

Although both ways are legitimate, in this paper we follow the second one. Given an evolution dynamics for the microstate probabilities, we first seek for an expedient formula for the entropic divergence [28]: a non-negative measure between two probability distributions (in a continuous model probability distribution functions (PDFs) which shrinks during the dynamical evolution. Possibly for two arbitrary initial distributions. Or at least between an arbitrary initial distribution and the stationary distribution. Once this convergence criterion is fulfilled, we study the entropic divergence from the uniform distribution. In special cases, we shall realize that the entropic divergence from the uniform distribution can be written as proportional to a difference of the same functionality on the uniform and on the investigated distribution. Finally this functional will be proposed as a candidate for an entropy formula. For linear master equations describing the evolution of probability, this turns out to be the Boltzmann entropy formula, while for other master equations, different formulas result. In particular, for a fractal power dependence on the probability, the Tsallis entropy formula emerges. Our method described above does not assume detailed balance while delivering the proof, in this way it is valid in a much broader class of dynamics than Boltzmann’s famous H-theorem.

It is often assumed that a non-exponential energy distribution would be the sign for non-extensive entropy. This assumption is, however, mistaken. Both a non-extensive entropy formula with a linear constraint on the energy and a traditional, logarithmic entropy formula with a constraint on another function of the energy lead to non-exponential PDFs in equilibrium. Moreover, several physical systems tend to a statistical behavior with a non-exponential stationary distribution without reaching equilibrium (open systems). In the framework of local growth and global reset model (LGGR) we demonstrate that far from the detailed balance state stationary distributions show a thermodynamical behavior, akin to the fluctuation–dissipation theorem [29]. Examples in income distribution, popularity measured by citations and hadron multiplicities in high-energy collisions demonstrate the flexibility and usefulness of the LGGR model.

As part of this presentation, we repeat some formulas published already earlier [26,28]. Our purpose is to provide the reader with a self-contained chain of thoughts, followable without further reading. For helping to concentrate on the distinction, however, we summarize here briefly the main development relative to our earlier publications. The general proof of deriving of the shrinking of the entropic divergence is now given for two arbitrary distributions, contrary to the former and more traditional derivation when one of the distributions is the stationary one. We also present some details on why, in the generalized Markovian dynamics, a similar proof based on the convexity property of the definition of the entropic divergence suffices only for an approach to the stationary distribution. Arguments for the generalization to nonlinear probability dynamics are shortly mentioned, but the applications in the framework of our LGGR model is again linear, concentrating on the processes with a stationary state without detailed balance.

## 2. Shrinking Entropic Divergence

Entropic divergence (sometimes cited as “entropic distance”, however, it is usually not a symmetric measure, nor does it satisfy any triangle inequality) is a non-negative real-valued functional of two probability distributions, ρ[P,Q]≥0. We also want that whenever this measure is zero, then the equality of the two distributions follows: $\rho \left[P,Q\right]=0\Rightarrow \;\forall n\;P_{n}=Q_{n}$. We add a further requirement, related to the evolution dynamics of the probabilities. This irreversibility property expresses that starting with an arbitrary distribution at time t=0, it will approach the stationary distribution, so their entropic divergence shrinks. To begin with we consider Csiszar’s f-divergence formula [30,31]
(1)$$\rho \left[P,Q\right]\;=\;\sum_{n}Q_{n}f\left(P_{n}/Q_{n}\right),$$
as a class of entropy divergence formulas investigated here. Applying the Jensen inequality [32,33] to this formula one obtains
(2)$$\rho \left[P,Q\right]\;\geq \;f\left(\sum_{n}P_{n}\right)\;=\;f\left(1\right)$$
for $f^{\prime \prime }>0$. The non-negativity property is hence satisfied by any function in the Csiszar formula with the correct second derivative and f(1)=0.

Next we review a linear dynamics for the probability evolution,
(3)$${\dot{P}}_{n}\;=\;\sum_{m}\;\left(w_{nm}P_{m}\;-\;w_{mn}P_{n}\right),$$
adjusted to conserve the normalization, i.e., $\sum_{n}{\dot{P}}_{n}=0$. Here the transition rates from the state *m* to *n*, $w_{nm}\geq 0$, define the probability evolution.

Now we are interested in the evolution of the entropic divergence, while both distributions—initially arbitrary—evolve governed by the common transition rates, $w_{nm}$. This description includes complex systems, where transitions can happen between non-neighboring, in the index space far away states, too. We make no restriction on n−m. Now using the definition in Equation (Equation 1) the evolution of the entropic divergence shows the following time derivative
(4)$$\dot{\rho }\;=\;\sum_{n}\left(\dot{P_{n}}f^{\prime }\left(\xi_{n}\right)\;+\;{\dot{Q}}_{n}\left[f\left(\xi_{n}\right)-\xi_{n}f^{\prime }\left(\xi_{n}\right)\right]\right),$$
with the notation $\xi_{n}=P_{n}/Q_{n}$. Substituting the dynamical master Equation (Equation 3) we obtain
(5)$$\dot{\rho }\;=\;\sum_{nm}\left(w_{nm}Q_{m}\left[\xi_{m}f^{\prime }\left(\xi_{n}\right)+f\left(\xi_{n}\right)-\xi_{n}f^{\prime }\left(\xi_{n}\right)\right]\;-\;w_{mn}Q_{n}\left[f(\xi_{n})\right]\right).$$


In the last term we exchange the summation indices to arrive at
(6)$$\dot{\rho }\;=\;\sum_{nm}w_{nm}Q_{m}\left[\left(\xi_{m}-\xi_{n}\right)f^{\prime }\left(\xi_{n}\right)+f\left(\xi_{n}\right)-f\left(\xi_{m}\right)\right].$$


Now a definite statement can be made about the sign of this expression by utilizing the Taylor series remainder theorem in the Lagrange form [34] for the yet unspecified function, f(ξ):
(7)$$f\left(\xi_{m}\right)\;=\;f\left(\xi_{n}\right)\;+\;\left(\xi_{m}-\xi_{n}\right)\;f^{\prime }\left(\xi_{n}\right)\;+\;\frac{1}{2}{\left(\xi_{m}-\xi_{n}\right)}^{2}\;f^{\prime \prime }\left(c_{nm}\right),$$
with $c_{nm}$ being an intermediate argument between $\xi_{n}$ and $\xi_{m}$.

Using this knowledge we finally obtain
(8)$$\dot{\rho }\;=\;-\frac{1}{2}\sum_{nm}w_{nm}Q_{m}{\left(\xi_{m}-\xi_{n}\right)}^{2}f^{\prime \prime }\left(c_{nm}\right).$$


For any definite sign of the second derivative of f(ξ), the above formula also has a definite sign, for $f^{\prime \prime }>0$ functions we have $\dot{\rho }\leq 0$. We note that $f^{\prime \prime }>0$ was the very same condition necessary for the non-negativity property of ρ itself, proven via the Jensen inequality, Equation (Equation 2). Consequently, we have proven that two arbitrary initial PDFs evolving via the linear master Equation (Equation 3) show a shrinking entropic f-divergence between them. This shrinking of the f-divergence from the stationary distribution to an arbitrary one is a basic property, discussed in the literature on Markov processes (for a review see e.g., [35]).

How far can this result be generalized for nonlinear evolution scenarios? We consider a slight modification using a positive function of the probability, a(P), in the master equation
(9)$${\dot{P}}_{n}\;=\;\sum_{m}\left(w_{nm}a\left(P_{m}\right)\;-\;w_{mn}a\left(P_{n}\right)\right).$$


Here a(P) can be any positive function, including the most known a(P)=P case. In some other complex system kinetics here powers may occur, as in chemical reaction networks [36], morphogenesis [37] or in analogy to some models of finance dynamics generalizing the Fokker-Planck equation to a nonlinear dependence on the phase space density [38]. Other functions may appear when simulating Boltzmann–Uehling–Uhlenbeck type of finite-state blocking or enhancement factors as in quantum transport models [39]. In such problems, one would apply a(P)=P/(1±P).

In this case, the above presented general proof does not apply. The more general trace formula for the entropic divergence,
(10)$$\rho \left[P,Q\right]\;=\;\sum_{n}\sigma \left(P_{n},Q_{n}\right),$$
has the rate of change
(11)$$\dot{\rho }\;=\;\sum_{n}\left({\dot{P}}_{n}\frac{\partial \sigma }{\partial P_{n}}\;+\;{\dot{Q}}_{n}\frac{\partial \sigma }{\partial Q_{n}}\right).$$


Replacing the dynamical Equation (Equation 9) both for ${\dot{P}}_{n}$ and ${\dot{Q}}_{n}$ and re-arranging the double summation indices as well as using the notation $\xi_{n}=a\left(P_{n}\right)/a\left(Q_{n}\right)$ we obtain
(12)$$\dot{\rho }\;=\;\sum_{n,m}w_{nm}a\left(Q_{m}\right)\left[\xi_{m}\left(\frac{\partial \sigma }{\partial P_{n}}-\frac{\partial \sigma }{\partial P_{m}}\right)\;+\;\left(\frac{\partial \sigma }{\partial Q_{n}}-\frac{\partial \sigma }{\partial Q_{m}}\right)\right].$$


For simplifying we introduce the definition
(13)$$f\left(\xi \right)\;\equiv \;\xi \frac{\partial \sigma }{\partial P}\;+\;\frac{\partial \sigma }{\partial Q}$$
for the arbitrary indexed quantities. Using this assertion the change of the divergence is straightforwardly written as
(14)$$\dot{\rho }\;=\;\sum_{n,m}w_{nm}a\left(Q_{m}\right)\left[\left(\xi_{m}-\xi_{n}\right)\frac{\partial \sigma }{\partial P_{n}}+f\left(\xi_{n}\right)-f\left(\xi_{m}\right)\right].$$


In order to apply the proof given in the linear master equation case and recited above, we have to achieve
(15)$$\frac{\partial \sigma }{\partial P_{n}}\;=\;f^{\prime }\left(\xi_{n}\right)\;\mathrm{and}\;\frac{\partial \sigma }{\partial Q_{n}}\;=\;f\left(\xi_{n}\right)\;-\;\xi_{n}f^{\prime }\left(\xi_{n}\right).$$


Whether it is possible or not, can be decided upon investigating the mixed second partial derivative of σ(P,Q), i.e., the integrability condition:
(16)$$\frac{\partial^{2}\sigma }{\partial Q_{n}\partial P_{n}}\;=\;\frac{\partial^{2}\sigma }{\partial P_{n}\partial Q_{n}}.$$


In our particular case this results in
(17)$$-\xi_{n}f^{\prime \prime }\left(\xi_{n}\right)\frac{a^{\prime }\left(Q_{n}\right)}{a(Q_{n})}\;=\;-\xi_{n}f^{\prime \prime }\left(\xi_{n}\right)\frac{a^{\prime }\left(P_{n}\right)}{a(Q_{n})}.$$


This does not happen, unless $a^{\prime }\left(P\right)=a^{\prime }\left(Q\right)$. This restricts us to the linear case or motivates investigations of non-trace like entropy divergence formulas.

It can be proven, however, that the above more general entropy divergence formula Equation (Equation 10), shrinks if we take $Q_{n}$ as the stationary distribution. The latter is defined by the ”total balance” condition
(18)$$0\;=\;\sum_{m}\left(w_{nm}a\left(Q_{m}\right)-w_{mn}a\left(Q_{n}\right)\right).$$


We note that the “detailed balance” condition would annulate each term in the above sum, not only the total result. That classical condition is actually a condition on the transition rates, reflecting microscopic time reversibility, while the total balance (consisting of n=1,2,…W equations) simply defines the stationary distribution $Q_{n}$.

In order to use the argumentation which holds in the linear case, one has only to satisfy
(19)$$\frac{\partial \sigma }{\partial P_{n}}\;=\;f^{\prime }\left(\xi_{n}\right),$$
with the modified definition $\xi_{n}=a\left(P_{n}\right)/a\left(Q_{n}\right)$.

The above equation can be solved for any positive a(P) function, particular solutions are set by the natural constraint ρ[Q,Q]=0. This result, stating that all distributions converge to the stationary one due to such a(P)-dynamics, does not mean that the entropic distance between two non-stationary distribution would continuously shrink during their evolution. They will all reach the stationary state and reduce their entropic divergence to zero only in the final stage.

It is traditional to use the function f(ξ)=−lnξ for generating the entropic divergence formula. Indeed in this case $f^{\prime }\left(\xi \right)=-1/\xi$ and $f^{\prime \prime }\left(\xi \right)=1/\xi^{2}>0$. The usual argumentation behind this choice is a further property, making the core function, f(ξ), in the Csiszar formula additive. For composite systems with subsystems (1) and (2) the probabilities namely factorize if statistical independence holds: $P_{n}\left(1⊕2\right)=P_{n}\left(1\right)P_{n}\left(2\right)$ and similarly for $Q_{n}$. This makes ξ(1⊕2)=ξ(1)ξ(2) and for f(1⊕2)=f(1)+f(2) with positive second derivative only this choice remains. Using this starting point the entropic divergence formula, Equation (Equation 1), and returning to the a(P)=P case specifies to the Kullback–Leibler definition [5,40]
(20)$$\rho^{\left(KL\right)}\left[P,Q\right]\;=\;\sum_{n}Q_{n}\ln\frac{Q_{n}}{P_{n}}.$$


This formula contains a so called “cross entropy” term and a pure entropy term associated to the PDF *Q*, but it is not yet unique what entropy formula would follow from this. To make this last step we utilize our concept of complexity: the entropic divergence of the uniform distribution, $U_{n}=1/W$ for n=1,2,…W from $Q_{n}$, i.e.,
(21)$$\rho^{\left(KL\right)}\left[U,Q\right]\;=\;\sum_{n=1}^{W}Q_{n}\ln\left(WQ_{n}\right)\;=\;\ln W\;+\;\sum_{n=1}^{W}Q_{n}\ln Q_{n},$$
equals to a difference, $\rho^{\left(KL\right)}\left[U,Q\right]=S^{\left(BG\right)}\left[U\right]\;-\;S^{\left(BG\right)}\left[Q\right]$, with the traditional definition of the Boltzmann–Gibbs entropy
(22)$$S^{\left(BG\right)}\left[Q\right]\;\equiv \;-\sum_{n}Q_{n}\ln Q_{n}.$$


In this case the Kullback–Leibler based complexity measure turns out to be a difference of the Boltzmann entropies, as suggested by the P-linear master equation dynamics as a proper measure. Its generalization, however, is not a difference of more general entropy formulas. The entropic divergence formula based on the corresponding probability evolution description in the master equation has to be the starting point, and its specification for a divergence from the uniform distribution is either a difference of the same formula at *U* and *Q*, or not.

As an example, we investigate a power-like nonlinearity, $a\left(P\right)=P^{\lambda }$. In this case, we do not use the Csiszar formula, but solve Equation (Equation 19) with the traditional logarithmic ansatz, f(ξ)=−lnξ:
(23)$$\frac{\partial \sigma }{\partial P_{n}}\;=\;f^{\prime }\left(\xi_{n}\right)\;=\;-\frac{Q_{n}^{\lambda }}{P_{n}^{\lambda }}.$$


Its particular solution fixing ρ[Q,Q]=0 reads as
(24)$$\rho^{\left(POW\right)}\left[P,Q\right]\;=\;\sum_{n}Q_{n}\ln_{\lambda }\frac{Q_{n}}{P_{n}},$$
with the so called deformed logarithm function [41,42,43] parametrized with λ:
(25)$$\ln_{\lambda }x\;\equiv \;\frac{1-x^{\lambda -1}}{1-\lambda }.$$


We note that starting with an f(ξ) function of a deformed logarithm as above, the result again has the same form, just the new λ is the product of the starting parameter ν in the deformation and the power in the master equation level non-linearity, $\lambda_{\mathrm{new}}=\lambda \nu$. The result Equation (Equation 24) is robust. Finally the suggested complexity measure becomes
(26)$$\rho^{\left(POW\right)}\left[U,Q\right]\;=\;\sum_{n=1}^{W}Q_{n}\ln_{\lambda }\left(WQ_{n}\right)\;=\;W^{\lambda -1}\left(S_{T}\left[U\right]-S_{T}\left[Q\right]\right).$$


It is proportional to a difference of Tsallis entropies,
(27)$$S_{T}\left[Q\right]\;\equiv \;-\sum_{n}Q_{n}\ln_{\lambda }Q_{n},\;S_{T}\left[U\right]\;=\;-\ln_{\lambda }\left(1/W\right),$$
and hence by construction proves that the Tsallis entropy is also maximal at the uniform distribution, since ρ≥0 holds, based on the Jensen inequality applied to Equation (Equation 24). The forefactor $W^{\lambda -1}$, depending on the number of states, signalizes non-extensivity in this case.

## 3. LGGR: A Linear Model for Local Growth and Global Resets

The master Equation (Equation 3) and its nonlinear generalization (Equation 9) describing the evolution of a probability distribution over possible physical microstates are quite general. The familiar diffusion problem in one dimension is described by transition rates between neighboring indices only, allowing to increase or decrease the index by one in a short time Δt. Their symmetric part is responsible for the diffusion coefficient while their difference provides a driving force for the drift. In higher dimensional diffusion problems there are also some sparse nonzero elements according to the finite size, like $w_{n+N_{x},n}$ for a local jump in the positive *y* direction, etc. These systems in the continuum limit are equivalent to a simple Fokker–Planck equation [44,45,46].

On the other hand in the up to date research we often face complex systems with underlying dynamics far from the detailed balance: the ratio of reversed micro-transition rates is far from looking as a ratio of a simple given function of indices, $w_{nm}/w_{mn}\neq Q_{n}/Q_{m}$. In this paper we present a particular simple model for extremely asymmetric processes: the growth rate is local in the index, while there is a global reset rate from anywhere to the starting (ground) state with index zero. In this local growth, global reset (LGGR) model we assume the following structure for the transition rate matrix (m→n):
(28)$$w_{nm}\;=\;\mu_{m}\delta_{n,m+1}\;+\;\gamma_{m}\delta_{n,0}.$$


The resulting linear master equation becomes
(29)$${\dot{P}}_{n}\;=\;\mu_{n-1}P_{n-1}\;-\;\left(\mu_{n}+\gamma_{n}\right)P_{n}$$
for n>0. The evolution for the n=0 state, ${\dot{P}}_{0}$, then includes the summation
(30)$${\dot{P}}_{0}\;=\;\sum_{m}\gamma_{m}P_{m}\;-\;\left(\mu_{0}+\gamma_{0}\right)P_{0},$$
but alternatively $P_{0}$ can be obtained from the normalization condition, $P_{0}=1-\sum_{n=1}^{\infty }P_{n}$. Now the complexity measure is given by
(31)$$\rho^{\left(LGGR\right)}\left[U,Q\right]\;=\;\sum_{n=1}^{W}Q_{n}f^{\left(KL\right)}\left(WQ_{n}\right)\;=\;\ln W+\sum_{n=1}^{W}Q_{n}\ln Q_{n},$$
and we are especially interested in its value for the stationary distribution of a given LGGR model. So the missing link is the computation of the stationary PDF in terms of $\mu_{n}$ and $\gamma_{n}$.

Here we review some of the most important LGGR models. The stationary distribution is obtained by the recursive resolution of the ${\dot{Q}}_{n}=0$ condition,
(32)$$Q_{n}\;=\;\frac{\mu_{n-1}}{\mu_{n}+\gamma_{n}}\;Q_{n-1}\;=\;Q_{0}\;\prod_{j=1}^{n}\frac{\mu_{j-1}}{\mu_{j}+\gamma_{j}}.$$


For constant growth and reset rates, $\mu_{n}=\sigma$, $\gamma_{n}=\gamma$, the result is the exponential distribution,
(33)$$Q_{n}\;=\;Q_{0}\;e^{-\beta n}$$
with β=ln(1+γ/σ). In this LGGR model with the Kullback–Leibler divergence and the traditional Boltzmannian entropy formula also non-exponential stationary distributions emerge if the transition rates are state-dependent. With a growth rate featuring linear preference, $\mu_{n}=\sigma \left(n+b\right)$, the stationary solution is a Waring distribution [47],
(34)$$Q_{n}\;=\;Q_{0}\;\frac{{\left(b\right)}_{n}}{{\left(c\right)}_{n}},$$
using the Pochhammer symbol ${\left(a\right)}_{n}=a\left(a+1\right)\ldots \left(a+n-1\right)$ and c=b+1+γ/σ. It exemplifies a power-law decay for large *n*, $Q_{n}∼n^{-1-\gamma /\sigma }$.

In the continuous LGGR version we consider the evolution equation
(35)$$\frac{\partial }{\partial t}P\left(x,t\right)\;=\;-\frac{\partial }{\partial x}\left(\mu (x)P(x,t)\right)-\gamma \left(x\right)P\left(x,t\right)\;+\;A\left(t\right)\delta \left(x\right)$$
with
(36)$$A\left(t\right)\;\equiv \;\int_{0}^{\infty }\gamma \left(x\right)P\left(x,t\right)\;dx\;-\;\mu \left(0\right)P\left(0,t\right).$$


This quantity may vanish for some special cases where μ(0)=0 and $\int_{0}^{\infty }\gamma \left(x\right)P\left(x,t\right)dx=0$. The corresponding stationary distribution becomes
(37)$$Q\left(x\right)\;=\;\frac{\mu (0)Q(0)}{\mu (x)}\;\exp\left(-\int_{0}^{x}\frac{\gamma (u)}{\mu (u)}\;du\right).$$


In particular the stationary PDF belonging to the linear local growth rate, μ(x)=σ(x+b), featuring linear preference and combined with a constant reset rate, γ(x)=γ, is given by the Tsallis–Pareto distribution:
(38)$$Q\left(x\right)\;=\;\frac{\gamma }{\sigma b}\;{\left(1+\frac{x}{b}\right)}^{-1-\gamma /\sigma }.$$


So power-law tailed distributions are stationary solutions to linear probability evolution equations with the traditional entropy and entropic divergence formula, if a constant global reset rate is connected with a linear preference in the local growth rate.

The total balance condition (Equation 37) determining the stationary PDF, on the other hand, can be viewed as a generalized fluctuation–dissipation theorem [48]. This form is useful for cases when the stationary PDF, Q(x), can be easily observed: than one of the rates provides information on the other. Simple integration of Equation (Equation 35) with $\dot{Q}\left(x\right)=0$ leads to such a relation, expressing the local growth rate with a distribution-tail average of the reset rate:
(39)$$\mu \left(x\right)\;=\;\frac{1}{Q(x)}\;\int_{x}^{\infty }\gamma \left(u\right)Q\left(u\right)\;du.$$


For an exponential PDF, $Q\left(x\right)=e^{-\beta x}/Z$, and constant reset rate, γ(x)=γ, one obtains μ=γ/β in analogy to the relation between diffusion rate and dissipation coefficient [49,50]. In general, for constant global reset rate γ, μ(x)/γ becomes the inverse hazard rate [51]. Such features transform LGGR to a simple and effective candidate for being an “Ising model”. It can be applied to far from equilibrium, with no detailed balance, and with complex system dynamics with various transition rates describing preferential scenarios.

## 4. Citation Popularity, Income Distribution and Some Further Application Areas

Finally we would like to review a few applications of this simplified LGGR treatment. As it has been published in [26], stationary PDF-s can be reproduced in several socio-economic or physical models. Socio-economic examples include popularity ranking from citation numbers [52,53,54], income [55,56,57] and settlement sizes [58]. For biological and ecological systems population abundance [59] on different levels (species, spatial, etc.) can be studied. For physical systems a good example is the hadron number distribution in high energy accelerator collision events [60].

In exceptional cases the income data are known on the level of individuals and are available for an extended period over several years. In such cases they allow for testing the growth and the entry/exit rates experimentally. We emphasize here that simply observing a certain distribution, Q(x), does not decide whether it is stationary, and even less whether an LGGR model could satisfactorily describe the real dynamics in its background. Whenever both observed distributions and transition rates with unidirected small-step growth and big-step global resets can be observed, then the LGGR can be verified or falsified.

As already discussed in earlier publications, citation popularity data follow a continuous Tsallis–Pareto distribution [52] which, scaled with the average number of citations, contains only one nontrivial fit parameter. In our analysis we have included Web of Science citations of some journals and institutes with international reputation as well as individuals. Furthermore Facebook likes and shares were counted, and similarly Youtube likes were recorded for diverse web pages. Surprisingly, all these popularity data collapse onto the same universal curve. The observed distributions can be understood by assuming linear preference in the local growth rate and a constant reset rate in the continuous version of the dynamics [52].

It is methodically useful to report here briefly about our recent investigations on income distribution. The starting point for our study is an individual level, complete income data set for a nine year period (2001–2009) in Cluj county (Romania) [61]. Beside the observed Q(x) PDF for income, we can follow the individual level income data to justify assumptions on the rates μ(x) and γ(x). Our preliminary findings include an improved ansatz on these rates. The reset rate, γ(x), occurs to be intelligent: at low income level it is negative, people enter into the income system mainly at this level, while at high income it is positive, the income is naturally high when they are close to retirement. The following fit describes well the averaged data
(40)$$\gamma \left(x\right)=K-\frac{b}{x+q},$$
with *K*, *b* and *q* freely adjustable parameters.

The local growth rate, μ(x), contains a simple linear term since the income (salary) increase is as a rule in per cent, so the amount of increase is proportional to the income at that moment. The assumed
(41)μ(x)=βx
functional form fits well the income data in Romania, Cluj district in the years 2001–2009.

For these growth and reset rates the resulting PDF is analytically obtained: using Equation (Equation 29), and normalizing it we obtain the Beta Prime distribution
(42)$$Q\left(x\right)=q^{\left(\frac{K}{\beta }-\frac{b}{\beta q}\right)}\frac{\Gamma \left(\frac{b}{q\beta }\right)}{\Gamma \left(\frac{b}{q\beta }-\frac{K}{\beta }\right)\Gamma \left(\frac{K}{\beta }\right)}{\left(1+\frac{x}{q}\right)}^{-\frac{b}{\beta q}}\;x^{\frac{b}{\beta q}-\frac{K}{\beta }-1}.$$


This formula, normalized and expressed in terms of z=x/〈x〉 contains only one parameter, and income distributions from Romania, Hungary, USA, Finland and Australia fall onto the common universal curve, $\langle x\rangle Q\left(x\right)=12z{\left(1+z\right)}^{-5}$.

Finally, let us mention the hadronization process in high energy elementary particle and heavy-ion collisions as a possible application field. Here, two distributions seem not to have a microscopic dynamical explanation, namely the multiplicity distribution of hadron number in a series of collisions with the same total energy, and the one-particle kinetic energy distribution averages over such collision events. The former seems to follow a negative binomial distribution (NBD) [62], although at the Large Hadron Collider (LHC) energies this is just a first approximation: the real distribution, including higher and higher kinetic energies in the count might require more sophisticated descriptions [63]. The latter on the other hand is beautifully fitted by a Tsallis–Pareto distribution in the mid-rapidity range, best interpolating between low, mid and higher transverse momenta in the range $p_{T}=0.8\ldots 12$ GeV [60,64].

We note that these distributions can be related to each other: a simple filling of a phase space volume whose dimensionality changes with the fluctuating number of produced hadrons event by event relates the NBD and the Tsallis–Pareto power-law tailed distribution in the individual kinetic energy. It is easy to demonstrate as follows.

In kinetic models, the microcanonical phase space is an energy shell in high dimension. The constant total energy constraint is a norm on the individual momenta, in a one-dimensional extreme relativistic jet this norm is $L_{1}$, $E=\sum_{i=1}^{n}\left|p_{i}\right|$, in a two-dimensional section at mid-rapidity of massive, non-relativistic particles on the other hand it is Pythagorean, $E=\sum_{i=1}^{2n}p_{i}^{2}/2m$. Both formulas are given for *n* particles. In a general asset we are interested in the surface of an *N*-ball with radius *R* in $L_{p}$ norm, which is known to be [65]
(43)$$V_{N}^{\left(p\right)}\left(R\right)\;=\;\frac{{\left[2R\;\Gamma (1+1/p)\right]}^{N}}{\Gamma (1+N/p)}.$$


One specifies N=n,p=1 and R(E)=E for the jets and N=2n,p=2 with $R\left(E\right)=\sqrt{2mE}$ for the two-dimensional ideal non-relativistic gas. The calculations deliver the results $V_{n}^{\left(1\right)}\left(E\right)={\left(2E\right)}^{n}/n!$ and $V_{2n}^{\left(2\right)}\left(\sqrt{2mE}\right)={\left(\pi mE\right)}^{n}/n!$. The microcanonical constraint volumes are obtained by a simple derivation from these formulas, $\Omega_{N}^{\left(p\right)}\left(E\right)=dV_{N}^{\left(p\right)}\left(R\left(E\right)\right)/dE$. Finally the phase space ratio (conditional probability) for a selected single particle having ε energy from this total of *E* in both cases is given by
(44)$$r_{n}^{\left(1\right)}\left(\varepsilon ,E\right)\;=\;r_{2n}^{\left(2\right)}\left(\varepsilon ,E\right)\;=\;\frac{\Omega (E-\varepsilon )}{\Omega (E)}\;=\;\frac{n-1}{E}{\left(1-\frac{\varepsilon }{E}\right)}^{n-2}.$$


This ratio is to be averaged over an NBD distribution of the newly produced n−2 hadrons (2 is minimally needed for making the collision),
(45)Qn=n+knfn(1+f)−n−k−1.


The result is a Tsallis–Pareto distribution [66] with some modifying factors:
(46)$$Q\left(\varepsilon \right)\;=\;\frac{1}{E}\left[1+f\left(k+1\right)-fk\frac{\varepsilon }{E}\right]{\left(1+f\frac{\varepsilon }{E}\right)}^{-k-2}.$$


For large 〈n〉≫1, characteristic for central heavy ion collisions, it approaches the classical form
(47)$$Q\left(\varepsilon \right)\;\approx \;\frac{1}{T}\;{\left(1+\frac{1}{k+1}\frac{\varepsilon }{T}\right)}^{-k-2},$$
with T=E/〈n〉=f(k+1), while for small 〈n〉≪1, more proper for describing elementary pp collisions, a similar Tsallis–Pareto law emerges with a modified power:
(48)$$Q\left(\varepsilon \right)\;\approx \;\frac{1}{E}\;{\left(1+\frac{1}{k+1}\frac{\varepsilon }{T}\right)}^{-k-1}.$$


Equation (Equation 46) represents a monotonic falling distribution in ε in the interval [0,E] and its integral over this interval is normalized to 1. Several measurements agree with this or similar functional forms, although some details can also be fitted with different forefactors to the Tsallis–Pareto term [67].

## 5. Summary

In conclusion, we presented an H-theorem like proof for stochastic evolution without detailed balance, in particular we have shown that Csiszár’s f-divergence formula for the entropic divergence shrinks between two arbitrary starting probability distributions for $f^{\prime \prime }>0$. The condition for this statement is that the same master equation, linear in the probability, governs the evolution of both. Further we presented the local growth global reset (LGGR) model, as a simple approach to processes far from the detailed balance and yet leading to interesting stationary distributions. This model utilizes only two rates, μ(x) for the growth and γ(x) for the reset to the x=0 point in the state space. A fluctuation-dissipation relation connects these rates via the normalized tail probability [68] of the stationary PDF, cf. Equation (Equation 39). Application fields for LGGR type approaches are numerous, here we examplified the citation popularity, the income distribution and the hadron distribution in high energy experiments.

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
