# Peer review of "Entropic Divergence and Entropy Related to Nonlinear Master Equations"

_entropy, 2019, doi:10.3390/e21100993_

Round 1

Reviewer 1 Report

See attached report.

Author Response

% Reply to the 3rd Report
% MDPI Entropy, 2019 SEP 28
%
% tsbiro 2019 OKT 01
%%%%%%%%%%%%%%%%%%%%%%%%%%%%%%%%%%%%%%%

Although the authors might have some interesting results, the present manuscript is, in
general, written in a very unclear and confusing way.

Response:
Actually we were intending to present a self-contained description, that readers
would not need to recheck a number of further publications; but naturally within
some page limit. Other critics is exactly about repeating a few derivations already
published elsewhere. The claim about the usability of the LGGR model is supported by
a number of published applications to real world data.
We have added some extra paragraphs in order to help the reader to follow the logic
of this presentation at various places.

-----------------------

The authors are specially sloppy when discussing the application to what they call the “local
growth and global reset (LGGR) model”. The authors first consider the discrete case of the
LGGR model, and then the continuous case, described by the partial differential equation
(28).

Now, the evolution equation (28) does not, in general, preserve the normalization of
the probability density P. As a consequence, in general, the evolution equation (28) cannot
have a normalized stationary solution. It can admit a formal stationary solution. But, in
general, this solution is not going to represent a legitimate normalized probability density.
Indeed, the formal solution (30) is not in general normalized, unless the problem is defined
on a restricted interval of the x variable. Now, if the formal solution is not normalized, it
makes no sense to evaluate entropic measures on it.

Response:
Of course, the presented solution is defined on the restricted interval x > 0.
We have now explicitly displayed the boundary flow term in eq.28 (new number 35)
- we were tacitly assuming - and by the virtue of this the probability normalization
is and was conserved.

------------------------

Reviewer 2 Report

This manuscript first studies and generalises an entropic divergence from the point of view of dynamics.
Assuming a general f-divergence of the relative entropy between two distributions, the authors show that this decreases with increasing time when the system evolves according to a standard Master equation. Next they perform a similar analysis on a generalised nonlinear master equation.
Further, a probabilistic model with growth and global resetting is discussed as a rather generic model for a variety of phenomena, namely citation popularity, income distribution, and hadron distribution.
The stationary distributions of these models are derived and shown to have a non exponential form.

I would like the authors to clarify the relation of this manuscript to their previous publication (reference 23). It seems that there is quite large overlap.
Also I believe that the shrinking of the f-divergence under time evolution is a well known result that can be found in text books on Markov chains. It should be adequately referenced.

Concerning the nonlinear generalisation of the master equation I think that some explanations are necessary. Such an equation cannot describe the time evolution of ordinary probabilities. So what would P represent in this case. In what, if any, contexts does such a generalisation appear?

It is not clear to me how the later part of the manuscript dealing with the local growth global resetting models is connected to the first part.

Author Response

%%%%%%%% Reply to Report1.txt / MDPI Entropy 2019 %%%%%%%%%%%%
%
% tsbiro 2019.OKT.01.
%%%%%%%%%%%%%%%%%%%%%%%%%%%%%%%%%%%%%%%%%%%%%%%%%%%%%%%%%%%%%%

I would like the authors to clarify the relation of this manuscript to their previous publication (reference 23).
It seems that there is quite large overlap.

Response:
Indeed the content overlaps somewhat. The improvements are twofold: i) the shrinking of f-divergence
is investigated now between two arbitrary distributions governed by the same master equation:
it succeeds in the linear case and does not succeed in the general nonlinear case. Previously
only the convergence to the stationary distribution was cared for (as it is usual).
ii) There are new results on investigating income distributions in Hungary, Romania, USA and Finland
for a decade. We briefly refer to these results in the context of our LGGR model.
We have inserted a paragraph on these issues at the end of section 1.
We have also set our approach into the context of research on log-Sobolev inequality improvements.
-------------------

Also I believe that the shrinking of the f-divergence under time evolution is a well known result that can be
found in text books on Markov chains. It should be adequately referenced.

Response:
It must be so, since it can be proven in a few lines. Nevertheless we are not aware of such citable
textbooks on Markov chains. Most of the mathematical literature cares for discrete time processes,
from which the continuous time version follows as a limiting case.
We cite a recent review paper on Markov chain dynamics.

--------------------------

Concerning the nonlinear generalisation of the master equation I think that some explanations are necessary.
Such an equation cannot describe the time evolution of ordinary probabilities. So what would P represent in this case.
In what, if any, contexts does such a generalisation appear?

Response:
Why cannot be a probability described by the generalized equation? As long as a(P) is a positive function, the sum
of all P_n-s is constant in time, and they never become negative. So the normalization sum_n P_n=1 is conserved,
its time derivative being zero.
Examples for such nonlinear dynamics are evolution equations for N_i/N number ratios in chemical kinetics or in other models:
the a(P) function is very often a power instead of being linear. Another hint into this direction is
Lisa Borland's generalization of the Fokker-Planck equation applied in econophysics. Other functions
than power law are exemplified in the Boltzmann-Uhling-Uehlenbeck models, which generate a(P)=P/(1+-P) forms.
We have inserted a short description of this background at the introduction of old eq.9.
In the revised version we discuss in detail why in the general case
only a convergence to the stationary distribution can be proven, instead of a proof for any two.

----------------------------

It is not clear to me how the later part of the manuscript dealing with the local growth global resetting models is
connected to the first part.

Response:
The description of applications of the LGGR model exemplifies processes far from detailed balance,
and provides an overview of some non-Boltzmannian and yet thermodynamical behavior in complex systems.
We have clarified our view in the introduction on the connection of the LGGR model - showing no detailed balance
while being linear in P - to the more general framework.

Reviewer 3 Report

The paper reverse engineer entropy formulas from entropic divergence. The paper  presented an H-theorem like proof for stochastic evolution without detailed balance and the local growth global reset (LGGR) model. Finally, the paper examplified the citation popularity, the income distribution and the hadron distribution in high energy experiments.

This work is interesting and the paper is well organized. Some major revisions are listed below for the further improvement.

In introduction, the authors point “Among those the log-Sobolev type is the closest to our approach.”, please authors give a brief introduction to log-Sobolev type. Besides, please authors explain the difference between log-Sobolev type and our approach. In abstract, page 2, the authors point “Not any formula will do”, please authors explain this sentences. Besides, will the formula proposed by authors do? In abstract, Page 2, the authors point “In this paper we follow the second way”. What is the second way. Besides, why not use the first way ? Please authors revise the introduction to better understand it . The writing should be improved in several parts, please check the full paper, such as: In page 7, line 118, “In many cases the experimental data are on individual level and available for an extended period” In page 5, line 90, “The master equation (3) and its pendant eq.(9)”. Please authors unify. In page 7, line 124, “As already discussed in earlier publications, citation popularity data follow a continuous” The references could be updated and some recent progress on divergence should be mentioned as follows: Song Y, Deng Y. Divergence Measure of Belief Function and Its Application in Data Fusion. IEEE Access, 2019, 7: 107465-107472. Song Y, Deng Y. A new method to measure the divergence in evidential sensor data fusion. International Journal of Distributed Sensor Networks,15.4 (2019): 1550147719841295. Fei L, Deng Y. A new divergence measure for basic probability assignment and its applications in extremely uncertain environments. International Journal of Intelligent Systems, 34.4 (2019): 584-600.

Author Response

%%%%%%%%%%%%%%%%%%%%%%%%%%%%%%%%%%%%%%%%%
%
% Reply2.txt/MDPI entropy report2
%
% tsbiro 2019. OKT. 02.
%%%%%%%%%%%%%%%%%%%%%%%%%%%%%%%%%%%%%%%%

In introduction, the authors point “Among those the log-Sobolev type is the closest to our approach.”,
please authors give a brief introduction to log-Sobolev type. Besides, please authors explain the difference
between log-Sobolev type and our approach.

Response:
We extended the first paragraph on the log-Sobolev approach listing some related math research.
One of the uses of this framework was always being that of proving generalizations of the H-theorem.
Others involve measure shrinking, information loss based on inequalities between integral forms.

The nonreversible situation is more involved than the reversible physics, in general.
Nonreversible (e.g. shrinking distance) processes with space dependent birth and death with
nonlinear rates and immediate reset, or as it is also called “resurrection”, are studied with
success only in particular cases and setting conditions different from ours.
See A. G. Pakes, Comm. Stat. Stoch. Models Volume 13, 1997 - Issue 2
or A. Chen, H. Zhang, K. Liu, and K. Rennolls, Adv. App. Prob. 36, 267 (2004).
and Li, J. & Liu, Z. Sci. China Math. (2011) 54: 1043.
For remarks on nonlinear systems but still with detailed balance the most recent one is
Gorban, A. N. Universal Lyapunov functions for non-linear reaction networks. 
Communications in Nonlinear Science and Numerical Simulation, 79, (2019) 104910.

However we do not intend to inflate our paper into this direction and therefore would like to
cease to include further citations out of this Pandora's box.

------------------------------

In abstract, page 2, the authors point “Not any formula will do”,
please authors explain this sentences. Besides, will the formula proposed by authors do?

Response:
The sentence after this did explain what are the desired properties
of an entropy formula. We have reformulated this heading sentence in paragraph 2.

------------------------------------------

In abstract, Page 2,
the authors point “In this paper we follow the second way”. What is the second way. Besides, why not use the first way ?
Please authors revise the introduction to better understand it .

Response:
We discuss in the preceding paragraph two possible approaches. Now we explicitly name them
"first" and "second". We added that both ways are legitimate, in the present essay we
just follow the second one.

----------------------------------------

The writing should be improved in several parts,
please check the full paper, such as: In page 7, line 118, “In many cases the experimental data are on individual
level and available for an extended period”

Response:
We have changed the sentence in section 4 paragraph 2.

-------------------------------------------

In page 5, line 90, “The master equation (3) and its pendant eq.(9)”.
Please authors unify.

Response:
We have reformulated 'pendant' to 'particular generalization'

----------------------------------------

In page 7, line 124, “As already discussed in earlier publications, citation popularity data
follow a continuous”

Response:
The word "data" is plural. We refer here to multiple and repeated data collections.
After eq.42 we inserted an explaining sentence.
More on this at
https://www.theguardian.com/news/datablog/2010/jul/16/data-plural-singular

----------------------------------

The references could be updated and some recent progress on divergence should be mentioned as
follows: Song Y, Deng Y. Divergence Measure of Belief Function and Its Application in Data Fusion. IEEE Access, 2019,
7: 107465-107472. Song Y, Deng Y. A new method to measure the divergence in evidential sensor data fusion.
International Journal of Distributed Sensor Networks,15.4 (2019): 1550147719841295. Fei L, Deng Y.
A new divergence measure for basic probability assignment and its applications in extremely uncertain environments.
International Journal of Intelligent Systems, 34.4 (2019): 584-600.

Response:
We have updated the references, and added some further citations.

Round 2

Reviewer 1 Report

The revised version of the manuscript can be published.

Reviewer 3 Report

The revised version well address my concerns. I recommend to publish it as is.